# Pathogenesis of Mucopolysaccharidoses, an Update

**DOI:** 10.3390/ijms21072515

**Published:** 2020-04-04

**Authors:** Simona Fecarotta, Antonietta Tarallo, Carla Damiano, Nadia Minopoli, Giancarlo Parenti

**Affiliations:** 1Department of Translational Medical Sciences, Federico II University, 80131 Naples, Italy; simona.fecarotta@unina.it (S.F.); tarallo@tigem.it (A.T.); c.damiano@tigem.it (C.D.); n.minopoli@tigem.it (N.M.); 2Telethon Institute of Genetics and Medicine, 80078 Pozzuoli, Italy

**Keywords:** mucopolysaccharidoses, LSDs, GAGs, autophagy, lysosomal storage disorders

## Abstract

The recent advancements in the knowledge of lysosomal biology and function have translated into an improved understanding of the pathophysiology of mucopolysaccharidoses (MPSs). The concept that MPS manifestations are direct consequences of lysosomal engorgement with undegraded glycosaminoglycans (GAGs) has been challenged by new information on the multiple biological roles of GAGs and by a new vision of the lysosome as a signaling hub involved in many critical cellular functions. MPS pathophysiology is now seen as the result of a complex cascade of secondary events that lead to dysfunction of several cellular processes and pathways, such as abnormal composition of membranes and its impact on vesicle fusion and trafficking; secondary storage of substrates; impairment of autophagy; impaired mitochondrial function and oxidative stress; dysregulation of signaling pathways. The characterization of this cascade of secondary cellular events is critical to better understand the pathophysiology of MPS clinical manifestations. In addition, some of these pathways may represent novel therapeutic targets and allow for the development of new therapies for these disorders.

## 1. Introduction

The knowledge on the pathophysiology of mucopolysaccharidoses (MPSs) has evolved over a long period of more than a century. The first MPSs were initially identified as distinct clinical entities [1,2,3,4,5] based on the description of their peculiar phenotypes and on the characterization of pathological findings, while the biochemistry, the biology, and the molecular bases of all these disorders continued to remain unknown for several decades. 

The discovery of lysosomes and the characterization of their role in cell biology and human disease marked a decisive turn in the history of MPSs [6]. Like for many other lysosomal storage diseases (LSDs), the pathophysiology of this class of disorders was ascribed to a block in the degradative function of lysosomes [7]. Accordingly, MPSs manifestations were viewed as direct consequences of lysosomal engorgement with undegraded mucopolysaccharides, also referred to as glycosaminoglycans (GAGs). 

Nowadays, with the changing vision of lysosomal biology, there is a renewed interest in the pathophysiology of LSDs, including MPSs. A large number of studies have challenged the traditional concept of lysosomes as exclusively degradative organelles, and LSDs are now viewed as disorders that simultaneously affect multiple cellular pathways and signaling cascades, each contributing to disease pathophysiology and to clinical manifestations.

## 2. Mucopolysaccharidoses (MPSs), Glycosaminoglycans (GAGs), and Lysosome Biology 

The MPSs are lysosomal storage diseases, due to deficiencies of enzymes involved in the breakdown of GAGs. GAGs are a heterogeneous family of highly sulfated, complex, linear polysaccharides that are composed of repeating disaccharide units and are present in every mammalian tissue [8]. The degradation of GAGs is performed by lysosomal hydrolases, either exoglycosidases that cleave the sugar residue at the end of the oligosaccharide chains, or sulfatases that remove sulfate from specific sugar residues [9]. These enzymes act sequentially; thus, a deficiency of one of these activities results in a block of further degradation of GAGs.

Different lines of research have changed our view of MPSs pathophysiology in recent years. Significant advances derived from studies on GAG functions in cell biology. GAGs have a variety of important biological roles. For a long time, they were thought to be exclusively constituents of extracellular matrix and membranes, and to be mainly involved in cell hydration and structural scaffolding. However, recent evidence indicates that GAGs also play a key role in cell signaling and modulate several biochemical processes that are fundamental for cell biology, including regulation of cell growth and proliferation, promotion of cell adhesion, anticoagulation, wound repair, and others [10]. In particular, extracellular GAGs represent reservoirs and co-receptors of different signaling molecules.

Additional and critical information on the pathophysiology of MPS was gathered thanks to studies that clarified the role of lysosomes in cell biology. The degradative function committed to the turnover of cellular constituents was long considered the primary task of lysosomes. This function was seen as a “house-keeping” process that was constitutively active in cells. Recent studies have challenged these assumptions and have provided incontrovertible evidence that the lysosomal compartment is part of a complex pathway, the autophagic-lysosomal pathway (ALP), and that lysosomal biogenesis and the activation of this pathway are tuned in adaptation to environmental stimuli, through the phosphorylation status and subcellular localization of transcription factor EB (TFEB) [11,12]. These activities are mainly regulated through the function of the multiprotein complex mammalian target of rapamycin complex 1 (mTORC1), localized at the cytosolic surface of the limiting membrane of the lysosome. mTORC1 is a kinase active on several substrates, including transcription factor EB (TFEB), that regulates ALP activation and recycling of cellular components, such as lipid stores [13]. Calcium signaling is also involved in regulating TFEB nuclear translocation and activation of autophagy [14]. Thanks to this characterization of lysosomal function and biology, lysosomes are now viewed as signaling hubs involved in many critical cellular processes, such as nutrient-sensing and regulation of metabolism, secretion, vesicle and membrane trafficking, growth, adaptive immunity, and others [11,15,16].

## 3. Lysosomal Storage and Secondary Disruption of Cellular Pathways

Given GAGs role in cell biology and in signaling pathways, it is not surprising that mutations causing defects in the degradation of these molecules have highly debilitating consequences, with multisystemic involvement and variable association of somatic, neurological, hematologic, and ocular symptoms. 

In addition, according to the newest information on lysosomal biology and on their central role in many cellular functions, multiple and diverse events are now emerging as important players in the pathogenesis of MPSs. Specifically, these events include storage of secondary substrates unrelated to the defective enzyme; abnormal composition of membranes and aberrant fusion and intracellular trafficking of vesicles; impairment of autophagy; mitochondrial dysfunction and oxidative stress; dysregulation of signaling pathways and activation of inflammation; abnormalities of calcium homeostasis and signaling [17,18,19,20] (Figure 1). It is likely that these factors or processes influence severity of symptoms and clinical manifestations, although, as this field is still largely unexplored, it is difficult to establish clear correlations between secondary cellular impairments and disease phenotypes [21].

### 3.1. Secondary Storage

In several LSDs, including MPSs, secondary storage of substrates that are not explained by the primary lysosomal defect has been consistently documented. The types of secondarily stored compounds are highly heterogeneous and include glycosphingolipids, phospholipids, and cholesterol [22,23,24,25,26,27].

Secondary storage has been described both in patients affected by MPSs and in animal models from different species. A comprehensive analysis of brain cortex tissues from post-mortem autopsy samples in patients affected with MPS I, II, IIIA, IIIC, and IIID was performed by biochemical analytical procedures (high performance liquid chromatography) and by histochemical staining of fixed tissues. This analysis revealed both changes in GAG composition (increase of heparan sulfate, decrease of keratan sulfate) and accumulation of secondary substrates including GM2 and GM3 gangliosides and lactosylceramide [28].

Similar studies were also done in animal models. Morphological and histological studies in huntaway dogs, a canine model of MPS IIIA, showed variably stained storage granules within neurons, including some that stained for gangliosides. On ultrastructural analysis, these granules contained both dense granular materials interpreted as GAGs, and a variety of multilamellar bodies interpreted as ganglioside accumulation [24].

In a feline model of MPS VI, abnormal lysosomal inclusions were pleiomorphic and resembled the zebra bodies and dense core inclusions typical of MPSs, while others appeared as membranous storage bodies characteristic of gangliosidoses. Pyramidal neurons showed positive staining for GM2 and GM3 gangliosides. In addition, positive filipin staining indicated storage of unesterified cholesterol [29]. As gangliosides can influence dendritogenesis during development, it is possible to speculate that accumulation of these compounds can trigger the changes in dendrite and axon morphology seen in MPS patients, leading to synaptic dysfunction, neuronal cell death in the brain, and neurodegeneration [30]. 

Secondary storage is not only confined to lysosomes but may contribute to accumulation in other compartments of toxic storage materials, including aggregated-prone proteins, alpha-synuclein, and damaged mitochondria [31] associated with common neurodegenerative disorders, such as Alzheimer’s, Parkinson’s, and Huntington’s disease. Alpha-synuclein accumulation has been consistently found in a variety of LSDs, including MPSs, and suggests correlations between alpha-synuclein aggregation toxicity and LSDs pathophysiology [32,33,34,35,36,37]. In MPS IIIA, a link between lysosomal dysfunction and presynaptic maintenance appeared to be mediated by a concurrent loss of α-synuclein and cysteine string protein α (CSPα) at nerve terminals. The relative loss of α-synuclein function by its abnormal autophagy was proposed as a contributing factor to neuronal degeneration [35].

The mechanisms leading to secondary storage are not clear. In principle, secondary storage may derive from inhibition by primary substrates of other lysosomal enzymes, from modification of the lysosomal environment, such as pH changes, or from impairment of vesicle trafficking through the endosomal/lysosomal system and the autophagic pathway [23,38,39]. 

While in the past secondary storage was typically considered a nonspecific and insignificant pathological feature of MPSs, newer studies support a substantial role of secondary storage as a major determinant in the pathophysiology of these disorders. 

### 3.2. Abnormal Composition of Membranes and Aberrant Intracellular Trafficking

An important consequence of secondary storage is its effect intracellular vesicle trafficking. Specifically, some substrates, such as cholesterol and other lipids, have been thought to play a role in altering membrane composition and jamming the endolysosomal system [23,38,39]. For example, the effect on vesicle trafficking has been studied in detail in the animal models of MPS type IIIA and of Multiple Sulfatase Deficiency, a peculiar disorder in which sulfatases, including those implicated in the breakdown of GAGs, are simultaneously deficient due to a defective post-translational modification of a cysteine at the catalytic site of these enzymes [40,41]. In the animal models of both diseases secondary cholesterol storage in endo-lysosomal membranes induces critical changes in the biochemistry and organization of lysosomal membranes. Abnormalities of the lipid composition of membranes impact on the function of a family of proteins, named soluble N-ethylmaleimide-sensitive factor attachment protein receptors (SNAREs), which are crucial components of the cellular membrane fusion machinery and are responsible for mediating membrane fusion processes in cells. Thus, SNAREs dysfunction leads to an impaired ability of lysosomal membranes to fuse with other membranes of other vesicles, such as endosomes and autophagosomes [42].

The defective trafficking of vesicles has several deleterious consequences and has been thought to have a role in the development of neuropathology in MPSs. For example, in cultured adrenal chromaffin cells from in MPS type IIIA, evidence of impaired exocytosis was observed [43]. It has been speculated that if these abnormalities also occur in central nervous system neurons, they may lead to a reduction in neurotransmitter release and explain some aspects of the MPS type IIIA neurological phenotype. 

In addition, recent data indicate that alpha-synuclein is a key chaperone assisting synaptic vesicle recycling and transmission at presynaptic terminals [44,45] by sustaining the function of the specific set of SNARE proteins involved in the synaptic vesicle trafficking [46,47], and that in the MPS type IIIA mouse model SNAREs dysfunction impairs synaptic vesicle recycling and neurotransmission, contributing to neurodegeneration [35].

### 3.3. Abnormal Autophagy

An important and deleterious effect of lysosomal storage and of defective vesicle trafficking is the impairment of the ALP and a block or reduction of the autophagic flux. Autophagy is an evolutionary conserved catabolic process that allows lysosomal delivery of intracellular components destined to degradation and turn over. A functional ALP is critical for many cell functions and for cell survival; thus, an impairment of this process may have disastrous consequences. This pathway has been shown to be affected in several LSDs, including MPSs, [33,48,49]. Original studies on the impairment of autophagy in LSDs were performed in animal models of an MPS IIIA and of multiple sulfatase deficiency. In both disorders, accumulation of immature autophagosomes, decreased co-localization of lysosomal and autophagic markers, increased ubiquitin levels and p62/SQSTM1-positive puncta, increased the number of mitochondria in different areas of brains and in neurons was interpreted as the consequence of defective autophagosome-lysosome fusion [33]. Evidence suggesting that a block of autophagy contributes to the MPS IIIA phenotype has been obtained using an MPS IIIA animal model in Drosophila. These MPS IIIA flies showed reduced ability to climb, indicating neurological impairment. Knockdown of Atg18 and Atg1, both essential components of the autophagic pathway, resulted in further worsening of performance in the climbing assay [50], suggesting a role of autophagy in the pathophysiology of the disease. Autophagy was shown to be also impaired in several other MPS animal models, including MPS II [51], MPS IIIC [52], MPS VI [53], and MPS VII [54]. Changes in expression of autophagy-related genes, coding for Atg1 and Atg18 proteins, were recently reported in MPS IIIA [50]. Another intriguing link between autophagy impairment and MPS has been discovered recently [55]. Mutations in the *VPS33A* gene, encoding a protein (VPS33A) that is involved in autophagy, resulted in an MPS-like disorder characterized by high levels of heparan sulfate in plasma and urine of patients, and in a phenotype sharing similarities with those of MPSs. 

The impairment of the autophagic flux has been recognized as important pathogenetic factor for neurodegeneration in lysosomal storage diseases, including MPSs [27,55]. In neurons, basal levels of autophagy are essential for neuronal function and survival, since they prevent toxic proteins from reaching harmful concentrations and contribute to the degradation of aged or damaged organelles, such as mitochondria [56,57]. A recent study in the mouse model of MPS IIIA has shown that restoration of the ALP was associated with reduced neuroinflammation and amelioration of memory deficits [37]. 

In addition, impairment of lysosome/autophagy pathway affects extracellular matrix formation and skeletal development and growth in MPSs [58,59].

Deregulation of mTORC1 signaling arrests bone growth in lysosomal storage disorders. In the mouse model of MPS VII lysosomal dysfunction induced a constitutive activation of mTORC1 in chondrocytes. As a consequence, chondrocytes fail to properly secrete collagens, the main components of the cartilage extracellular matrix. Rescue of the autophagy flux resulted in restored collagen levels in cartilage and ameliorated the bone phenotype [54].

### 3.4. Mitochondrial Dysfunction

A primary function of autophagy is to mediate mitochondrial turnover [60]. This selective form of autophagy is known as mitophagy and its importance in preserving functional integrity of mitochondria has been increasingly recognized in the past few years. Thus, it is not surprising that disorders characterized by an impairment of autophagy are associated with mitochondrial dysfunction, and that mitochondrial dysfunction contributes to the pathophysiology of these disorders. 

Perturbations in mitochondrial function and homeostasis caused by impaired autophagy have been recognized in several LSDs, including some MPSs [61], and have been proposed as one of the mechanisms underlying neurodegeneration [25,62,63,64]. Studies in an MPS IIIB mice have shown increased amounts of the small mitochondrial protein subunit C of the mitochondrial ATP synthase in specific brain regions, including the entorhinal and the somatosensory cortex. This finding was already evident at 1 month of age and was shown to increase with time [65]. Similar findings were observed post-mortem in the cerebral limbic system and central gray matter of the mid brain and pons in a patient with Hurler-Scheie MPS I [66].

Pathological findings have been described in detail in the murine model of MPS IIIC, in which accumulation of pleomorphic, swollen mitochondria containing disorganized or reduced cristae, was observed in neurons in all parts of the brain. These abnormalities appeared to be progressive. Some neurons containing swollen mitochondria were already detectable at 5 months of age, while by the age of 12 months mitochondrial damage was seen in the majority of neurons [52,63].

### 3.5. Oxidative Stress 

Increased oxidative stress and enhanced susceptibility of cells to mitochondria-mediated apoptotic insults are obvious consequences of defects in mitophagy and mitochondrial dysfunction. Elevation of reactive oxygen species (ROS) and accumulation of damaged mitochondria has been observed in some MPSs [67].

Some studies performed in animal models have linked oxidative stress to MPSs [68,69]. These studies showed the presence of oxidative stress already in the early stages of disease progression in the murine model of MPS IIIB. Other studies demonstrated an oxidative imbalance in animal models of MPS I [70] and MPS IIIA [71]. Elevated oxidative stress has also been documented in blood samples from patients affected by MPS type I [72] and MPS II [73]. In these patients, oxidative damage to proteins and lipids increased catalase activity and reduced total antioxidant status were found. Interestingly, oxidative stress may be directly linked and explain the activation of inflammation and the abnormalities of autophagy [74]. Studies in an MPS IIIB murine model suggest that oxidative stress is not a consequence, but a cause of neuroinflammation, since it is present at a very early stage in the brain [75].

### 3.6. Alteration of Signaling Pathways 

Non-physiologic activation of signaling cascades is an important and intriguing aspect of MPS pathophysiology that in recent years is attracting growing interest. Indeed, aberrant signaling may be directly implicated in the pathophysiology of some of the most prominent and most debilitating clinical manifestations of these disorders, such as pain, physical disability, neurodegeneration, skeletal abnormalities, and heart involvement. 

Several factors contribute to signaling dysregulation. One of them is the synthesis of aberrant GAGs that interferes with normal GAG interactions with different receptors, such as the fibroblast growth factors (FGFs), and with morphogens such as those implicated in neurogenesis, axonal guidance, and synaptogenesis. In addition, several of the secondary events triggered by storage (impaired autophagy, mitochondrial dysfunction and oxidative stress abnormal trafficking of vesicles, membranes, and membrane proteins) may affect the internalization and trafficking of signaling molecules [76]. Neurodegeneration and skeletal involvement are important examples of MPSs clinical manifestations that may be linked to altered signaling (Figure 2).

Severe neuroinflammation is a consistent finding in MPS animal models and may be a factor implicated in the progression of neurodegeneration. Accumulation of GAG-related oligosaccharides in microglia, likely released by lysosomal exocytosis, are thought to induce inflammation in the brain by activating toll-like receptor (TLR) receptors of microglia cells, and to induce release of inflammatory cytokines. Evidence of neuroinflammation in MPSs has been obtained both in vitro and in vivo. In vitro lipopolysaccharide (LPS)-TLR4 activated microglia has been shown to express and secrete inflammatory cytokines and chemokines such as tumor necrosis factor-α (TNFα), interleukin-1β (IL-1β), interleukin-6 (IL-6), and macrophage inflammatory chemochine ligand-3 (CCL3) [77]. In in vivo studies, an increase of activated astrocytes and microglial cells has been observed in the brain, in the cortical area, and in the spinal cord of several murine models of MPSs, such as MPS I, IIIA, IIIB, and IIIC [63,77,78]. In brains from the MPS IIIB mouse, a massive upregulation and activation of astrocytes and microglia, and secretion of inflammatory cytokines and other proteins related to immunity and macrophage function has been described early in disease progression [79]. The same mouse model showed activation TLR4/myeloid differentiation primary response 88 (MyD88) pathway in the brain [80]. In brains from the mouse model of MPS VII, a gene expression profile analysis revealed up-regulation of genes related to the immune system and inflammation. The patterns of gene expression dysregulation appeared specific for different brain regions, suggesting that specific brain regions may be more vulnerable to activation of inflammation than others [81].

The characterization of the molecular pathways underlying neuroinflammation in MPSs has possible therapeutic implications. The TLR4-TNFa pathway has been recognized as a potential therapeutic target. Preclinical studies with pentosan-sulfate, an FDA-approved drug with anti-inflammatory and pro-chondrogenic properties, showed clinical improvements in MPS VI rats and in MPS I dogs, with a reduction of pro-inflammatory cytokines in tissues and in the cerebrospinal fluid [82,83,84]. A pilot clinical study based on weekly pentosan-sulfate injections for 12 weeks in three male Japanese patients with attenuated MPS II resulted in decreased inflammatory cytokines macrophage migration inhibitory factor (MIF) and TNF-α [85].

Aberrant signaling also plays an important role in the pathophysiology of bone and skeletal involvement in MPSs. In this case excess of extracellular, rather than intracellular GAGs, appear to be most significant. In fact, chondrogenesis, the earliest phase of skeletal formation, is mostly controlled by cellular interactions between the extracellular matrix (of which GAGs are important components) and differentiation factors, other signaling molecules and transcription factors [86]. Recent studies point to a role of autophagy as quality control pathway of collagen, another important component of extracellular matrix. These studies suggest that an impairment of autophagy leads to a collagen proteostatic defects, thus providing a possible mechanism implicated in skeletal defects in LSDs [59].

Much attention has been paid to fibroblast growth factors (FGFs) signaling pathway. FGFs are a cytokine family that modulates cell growth, migration, differentiation, and neuroectodermal development [87]. Abnormally accumulated GAGs and defective proteoglycan desulfation have been shown to affect FGF2-heparan sulfate interactions and FGF signaling in the murine model of Multiple Sulphatase Deficiency. [33] and in multipotent adult progenitor cells derived from MPS I patients [88]. Exogenous and endogenous GAGs were also shown to modulate the bone morphogenetic protein-4 (BMP-4) signaling activity in MPS I cells [88]. Dysregulated FGF2 signaling was found in MPS I chondrocytes, together with altered GAG, FGF2, and Indian hedgehog distribution in growth plates from MPS I mice [89]. In two different MPS II animal models, D. rerio and M. musculus the FGF pathway activity was shown to be impaired during early stages of bone development. In both models, the FGF signaling deregulation anticipated a slow but progressive defect in bone differentiation [90]. Abnormally accumulated GAGs and defective proteoglycan desulfation have been shown to alter FGF2-heparan sulfate interactions and fibroblast FGF signaling pathway also in the murine model of multiple sulfatase deficiency [33].

Studies performed in the MPS VII canine model showed failed initiation of secondary ossification in vertebrae and long bones at the appropriate postnatal developmental stage and suggested dysregulation of signaling pathways modulating bone development and ossification. Epiphyseal chondrocytes showed abnormal persistence of Sox9 protein and were unable to successfully transition from proliferation to hypertrophy [91]. Targeted gene expression profiling showed differential expression of a number of genes involved in pathways important for the regulation of endochondral ossification. Osteoactivin (GPNMB) was the top upregulated gene. In addition, elements of key osteogenic pathways such as Wnt/β-catenin and BMP signaling were not upregulated in MPS VII during critical developmental window suggesting that these bone formation pathways are not activated [92].

Proteomic studies in the murine MPS I model revealed significant decreases in key structural and signaling extracellular matrix proteins, such as biglycan, fibromodulin, PRELP, type I collagen, lactotransferrin, and SERPINF1. Genome-wide expression analysis in the same mouse model identified several significantly dysregulated mRNAs (Adamts12, Aspn, Chad, Col2a1, Col9a1, Hapln4, Lum, Matn1, Mmp3, Ogn, Omd, P4ha2, Prelp, and Rab32) [93].

It has also been suggested that in MPS II perturbations of GAG catabolism may affect morphogens release and activity, such as sonic hedgehog (Shh) distribution and signaling. Uncleared GAGs may interfere extracellularly with Shh binding to Patched, therefore blocking Shh pathway transduction [51]. Dysregulation of the Shh and Wnt/β-catenin signaling has been linked to aberrant heart development and atrioventricular valve formation in a zebrafish mode of this disorder [94].

Impaired calcium homeostasis and signaling has been demonstrated in some LSDs, such as Niemann-Pick disease type C [95,96]. These abnormalities are of particular interest as calcium signaling is an essential process in cells and is maintained by the concerted action of channels, pumps, transporters, and receptors that maintain intracellular calcium stores. In recent years, lysosomes have emerged as a major intracellular calcium storage organelle, with an increasing role in triggering or modulating cellular functions such as endocytosis, calcium release from cellular organelles and autophagy. Evidence of disruption of calcium and proton homeostasis was also shown in MPS I [97]. 

## 4. Conclusions

The characterization of the cellular processes that are involved in the pathophysiology of lysosomal diseases has important implications for the treatment of LSDs, including MPSs. Some of the pathways that are dysregulated in these disorders may be pharmacologically or genetically manipulated and may represent novel and promising therapeutic targets. It is reasonable to think that future research will focus on these aspects, possibly developing complementary strategies to improve the outcome of traditional therapies aimed at restoring the function of defective enzymes.

## Figures and Tables

**Figure 1 ijms-21-02515-f001:**
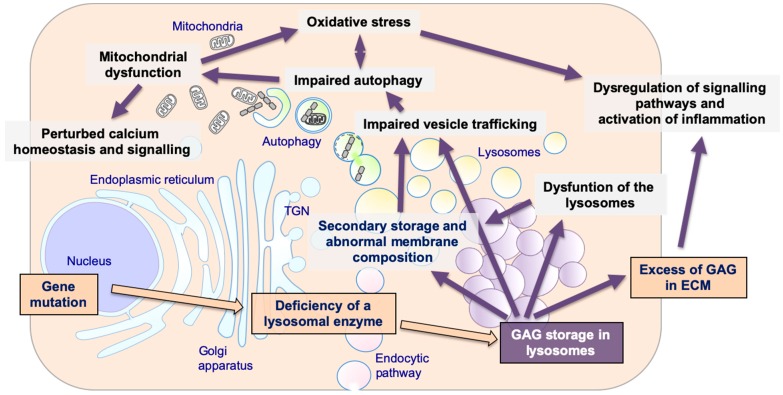
The pathogenetic cascade of mucopolysaccharidoses (MPSs). Multiple pathogenetic events in MPSs, including primary storage of glycosaminoglycans (GAGs) and secondary disrupted pathways: secondary storage of different substrates and abnormal membranes composition; impaired fusion and vesicles trafficking; impairment of autophagy; mitochondrial dysfunction and oxidative stress; dysregulation of signaling pathways and activation of inflammation; impaired calcium homeostasis and signaling.

**Figure 2 ijms-21-02515-f002:**
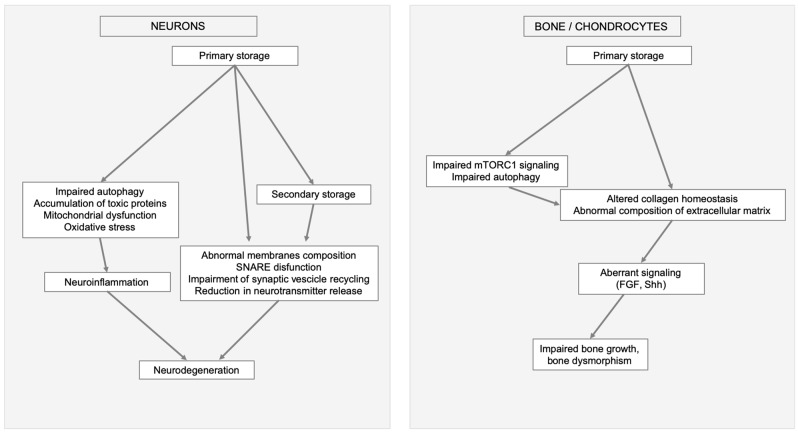
Examples of MPSs clinical manifestation triggered by primary storage and linked to secondary cellular impairments and altered signaling. In neurons and brain (left) impaired autophagy, mitochondrial dysfunction, oxidative stress, and secondary storage cause neuroinflammation, abnormal membranes composition, soluble N-ethylmaleimide-sensitive factor attachment protein receptors (SNARE) disfunction, impairment of synaptic vesicle recycling, and reduction in neurotransmitter release ultimately leading to neurodegeneration. In bone (right) secondary events, like aberrant mammalian target of rapamycin complex 1 (mTORC1) signaling, impaired autophagy, altered collagen homeostasis, abnormal composition of extracellular matrix e aberrant signaling, result in impaired bone growth and bone dysmorphism.

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
