# Peer review of "Pathogenesis of Mucopolysaccharidoses, an Update"

_ijms, 2020, doi:10.3390/ijms21072515_

Round 1

Reviewer 1 Report

To the authors:

The manuscript entitled "Pathogenesis of mucopolysaccharidoses, an update" by Simona Fecarotta, et al.is a comprehensive review for mucopolysaccharidoses (MPS), especially focused on the pathophysiology of MPS to lysosomal biology. Generally this manuscript is well summarized and written. Reviewer has some minor suggestions which may increase the significance of the manuscript, but all of them may not be possible to be addressed.

Authors describe detailed cellular phenotypes of MPS, however explanation how these cellular phenotypes link to (or suggested to contribute to) systemic symptoms, such as bone deformations or neurological disorders, observed in MPS patients might not be enough. Since the title of this manuscript is "Pathogenesis of MPS", are there some more suggestions for the relations between cellular impairment and such clinical phenotypes?

Cellular phenotypes written in the sections 3 are informative, which clearly explain what are occurring within disease cells. Most of these cellular phenotypes are commonly observed in LSD. Among them, which is relatively MPS specific cellular phenotypes, or which is explained with GAG accumulation?

This manuscript still contains minor English mistakes, abbreviation errors, and unformatted descriptions, which should be revised during the editorial process.

Author Response

Response to Reviewer 1 Comments 

Point 1: The manuscript entitled "Pathogenesis of mucopolysaccharidoses, an update" by Simona Fecarotta, et al. is a comprehensive review for mucopolysaccharidoses (MPS), especially focused on the pathophysiology of MPS to lysosomal biology. Generally this manuscript is well summarized and written. Reviewer has some minor suggestions which may increase the significance of the manuscript, but all of them may not be possible to be addressed.

Authors describe detailed cellular phenotypes of MPS, however explanation how these cellular phenotypes link to (or suggested to contribute to) systemic symptoms, such as bone deformations or neurological disorders, observed in MPS patients might not be enough. Since the title of this manuscript is "Pathogenesis of MPS", are there some more suggestions for the relations between cellular impairment and such clinical phenotypes?

Response 1: We have already given in the original manuscript some information on phenotypic correlations and related references, such as:

- intracellular trafficking alterations may lead to defects of neurotransmission and contributing to neurological phenotypes and neurodegeneration (section 3.2.);

- problems related to abnormal autophagy (section 3.3.) may lead to neuroinflammation and memory deficits or arrest of bone growth;

- specific alteration of signaling pathways described in section 3.6. may lead to neurodegeneration, skeletal involvement, failure in secondary ossification or aberrant heart development and atrioventricular valve formation.

In the revised manuscript we have now provided some additional comments on this point, as suggested by reviewer

  • It is likely that these factors or processes influence severity of symptoms and clinical manifestations, although, being this field still largely unexplored, it is difficult to establish clear correlations between secondary cellular impairments and disease phenotypes. (lines 85-88)

  • In MPS IIIA a link between lysosomal dysfunction and presynaptic maintenance appeared to be mediated by a concurrent loss of α‐synuclein and cysteine string protein α (CSPα) at nerve terminals. The relative loss of α‐synuclein function by its abnormal autophagy was proposed as a contributing factor to neuronal degeneration (lines 126-129) [Ref 33 Sambri et al]

  • As gangliosides can influence dendritogenesis during development, it is possible to speculate that accumulation of these compounds can trigger the changes in dendrite and axon morphology seen in MPS patients, leading to synaptic dysfunction, neuronal cell death in the brain and neurodegeneration (lines116-119);

  • Studies in an MPS IIIB murine model suggest that oxidative stress in not a consequence, but a cause of neuroinflammation, since it is present at a very early stage in brain. (lines 234-236)

  • Recent studies point to a role of autophagy as quality control pathway of collagen, another important component of extracellular matrix. These studies suggest that an impairment of autophagy leads to a collagen proteostatic defects, thus providing a possible mechanism implicated in skeletal defects in LSDs (lines 291-294)

  • Abnormally accumulated GAGs and defective proteoglycan desulfation have been shown to alter FGF2-heparan sulfate interactions and fibroblast FGF signaling pathway also in the murine model of multiple sulfatase. (lines 305-307)

Point 2: Cellular phenotypes written in the sections 3 are informative, which clearly explain what are occurring within disease cells. Most of these cellular phenotypes are commonly observed in LSD. Among them, which is relatively MPS specific cellular phenotypes, or which is explained with GAG accumulation?

Response 2: Secondary disruptions of cellular pathways described are common emerging features of most LSDs, including MPSs (impaired autophagy, mitochondrial dysfunction and oxidative stress, abnormal trafficking of vesicles, membranes and membrane proteins). We have mentioned some peculiar MPSs’ aspect, such as:

  • As gangliosides can influence dendritogenesis during development, it is possible to speculate that accumulation of these compounds can trigger the changes in dendrite and axon morphology seen in MPS patients, leading to synaptic dysfunction, neuronal cell death in the brain and neurodegeneration (lines 116-119);
  • signaling dysregulation due to the synthesis of aberrant GAGs that interfere with normal GAG interactions with different receptors, such as the fibroblast growth factors (FGFs), and with morphogens such as those implicated in neurogenesis, axonal guidance and synaptogenesis (already described in section 3.6.)

Reviewer 2 Report

The authors have presented interesting review paper “Pathogenesis of mucopolysaccharidoses, update”. for the International Joural of Molecular Sciences. The author summarizes recent findings on the pathogenesis of mucopolysaccharidosis very well.

1 The author summarizes the various etiologies well, but if possible, can it be presented to the reader clearly by inserting some figures, especially in the pathology of the brain and bones?

Author Response

Response to Reviewer 1 Comments

Point 1: The authors have presented interesting review paper “Pathogenesis of mucopolysaccharidoses, update”. for the International Journal of Molecular Sciences. The author summarizes recent findings on the pathogenesis of mucopolysaccharidosis very well.

The author summarizes the various etiologies well, but if possible, can it be presented to the reader clearly by inserting some figures, especially in the pathology of the brain and bones?

Response 1: We have added an additional figure (Figure 2) to outline some aspects of brain and bones pathophysiology, as suggested. The figure is mentioned in the text (line 250).
